# The Association of Oxidative and Antioxidant Potential with Cardiometabolic Risk Profile in the Group of 60- to 65-Year-Old Seniors from Central Poland

**DOI:** 10.3390/antiox11061065

**Published:** 2022-05-27

**Authors:** Bartłomiej K. Sołtysik, Kamil Karolczak, Cezary Watała, Tomasz Kostka

**Affiliations:** 1Department of Geriatrics, Medical University of Lodz, 90-419 Łódź, Poland; tomasz.kostka@office365.umed.pl; 2Department of Haemostatic Disorders, Medical University of Lodz, 90-419 Łódź, Poland; kamil.karolczak@office365.umed.pl (K.K.); cezary.watala@office365.umed.pl (C.W.)

**Keywords:** oxidative stress, antioxidant status, cardiovascular risk, elderly

## Abstract

Pathogenesis of cardiovascular diseases is caused by, inter alia, oxidative stress. On the other hand, cardiovascular risk factors may cause redox imbalance. The pathological pathways between those components are to be determined. In the group comprised of 300 sex-matched subjects, we evaluated a number of cardiovascular risk factors: blood pressure, body mass, lipids, glucose, homocysteine, uric acid, von Willebrand factor (vWF), VCAM-1 and ICAM-1. The presence of cardiovascular diseases and drugs for their treatment were examined. Secondly, we assessed total antioxidative status (TAS), total oxidative status (TOS) and other markers of oxidative stress. TAS was inversely related to LDL cholesterol. TOS was positively associated with BMI and female sex, but negatively associated with the use of angiotensin II receptor antagonists. Plasma lipid peroxides concentration was positively related to ICAM-1 and presence of stroke, whereas platelet lipid peroxides were positively associated with vWF. Platelets proteins thiol groups were in a positive relationship with vWF, but in a negative relationship with uric acid and diagnosed lipid disorders. Both free thiol and amino groups were positively associated with plasma glucose. Platelets free amino groups were related to platelets count. Superoxide generation by blood platelets (both with and without homocysteine) was positively connected to glucose level. Among women, oxidative markers appear to be more related to glucose level, whereas among men they are related to body mass indices. TAS, TOS and oxidative markers are largely related to modifiable cardiovascular risk factors such as body mass, and intake of drugs such as angiotensin II receptor blockers. Plasma and platelet oxidation markers appear to be especially associated with glucose concentration. The presented analyses unanimously indicate strong connections between cardiovascular risk factors and redox potential and specify how cardiometabolic interventions may counter-balance oxidative stress.

## 1. Introduction

The pathogenesis of circulatory diseases is increasingly often sought in oxidative stress, especially as cardiovascular risk factors may also ignite redox imbalance. Some factors, such as hyperglycemia or increased blood pressure, may disturb redox status, and the oxidative outcomes may hamper the control of cardiometabolic disorders. This is especially true in the older population, in which this imbalance is further modified by the agents prescribed for their treatment [1,2,3]. Thus, research on the pleiotropic role of oxidative stress and antioxidant potential in the pathogenesis of cardiovascular diseases, especially among older subjects, is still very essential.

The markers of oxidative stress arise in a situation of dysregulation between reactive oxygen species (ROS) generation and neutralization. Antioxidative protective mechanisms may be tracked by the total antioxidative status (TAS) of body fluids, including blood plasma. TAS depends primarily on the body’s own resources (the concentration of endogenous substances with antioxidant potential: bilirubin, uric acid, glutathione) [4,5,6,7]. The substances with reducing properties supplied with food also influence the antioxidant capacity, i.e., vitamins C, E and β-carotene. Many compounds exhibit antioxidant synergism, e.g., glutathione regenerates ascorbate, and ascorbate in turn reproduces α-tocopherol [8,9]. The interaction between antioxidants results in a greater antioxidant capacity than the action of each compound individually. The TAS value may be an indicator of the body’s protective ability against oxidation, a measure of the antioxidant reserve, that inhibits oxidation processes occurring in the course of many diseases and conditions, such as smoking and obesity. In addition to the well-known endogenous and dietetic antioxidants, to the group of factors with such an effect, one can add statins, angiotensin converting enzyme inhibitors and angiotensin II receptor blockers (ARB) [10].

The total oxidative activity is defined by the total oxidative plasma status (TOS). This parameter allows to determine the plasma ability to produce free radicals. However, this process may also be reflected in reactions such as lipid peroxidation, oxidation of protein thiol and amino groups or superoxide anion generation. These processes activate blood platelets and may lead to thrombosis, vascular remodeling, vasoconstriction and inflammation [11]. Notwithstanding, prolonged high level of oxidative stress may induce endogenous antioxidant defense mechanisms, which may paradoxically, by up-regulation, lead to improvement of health and longevity [12]. Oxidative changes in proteins disrupt the cell functioning, leading ultimately to apoptosis. Oxidation of protein thiol groups (-SH) by superoxide radical, hydrogen peroxide or hydroxide radical leads to biological inactivation of proteins, oxidation of membrane thiol groups and cell barrier disintegration. Various oxidative reactions may be modulated into a uniform effect in the form of oxidation of cellular structure.

Currently, the observation of cardiometabolic risk requires the differentiated approach and assessment of not only well-known classical cardiovascular risk factors, but also non-classical ones. Among those which negatively influence cardiovascular risk is von Willebrand factor (vWF). The concentration of this protein is positively related to arterial thrombosis, myocardial infarction and ischemic stroke [13]. A similar situation can be assessed for several markers of inflammation, which have been acknowledged as potential cardiovascular risk triggers. Among them are cell adhesion molecules such as intercellular adhesion molecule 1 (ICAM-1) and vascular cell adhesion protein-1 (VCAM-1) [14,15]. The above-mentioned data lead to the inclusion of those risk factors in the study, which allow to investigate the linkage between oxidative stress and cardiometabolic risk in an innovative way.

Up until now, the relationship between a wide range of classical and non-classical cardiometabolic risk factors versus TAS, TOS, and a broad spectrum of oxidative stress markers has not been systematically analyzed in the elderly population. The aim of this study was to estimate which of the most recognizable cardiovascular risk factors, cardiovascular disorders and cardiometabolic drugs may play crucial role in the oxide-reducing imbalance. In particular, we wanted to verify whether any previously non-examined elements of the oxide-reducing system might have a special link to any of cardiometabolic risk factors.

## 2. Materials and Methods

### 2.1. Study Design and Subjects Recruitment

The paper presents the results of cross-sectional study obtained in the project entitled “The occurrence of oxidative stress and selected factors for cardiovascular risk and functional status of older people in the context of workload” (funded by the Central Institute For Labour Protection-National Research Institute, Warsaw, Poland). Two elementary inclusion criteria were the age within the range of 60 to 65 years and the subjects’ willingness to participate in the study.

All steps of human experimentation were conducted under the guidelines of the Declaration of Helsinki for human research. The study was approved by the Committee on the Ethics of Research of Human Experimentation at the Medical University of Lodz, Poland. Written abstract of the experiment, including detailed information regarding the study objectives, design, risks, and benefits was given to each of the participants. At the very beginning of the study, the respondents obtained information about the purpose, course and use of possible results of the research, as well as the opportunity to refuse further participation in the study at any stage of the probe, without providing the reason for this decision. The study group initially comprised of 350 sex-matched participants. To further the examination, 150 women and 150 men were classified accordingly.

Weight and height measurements were taken from barefooted participants, and waist and hip circumference were measured; body mass index (BMI), waist–hip ratio (WHR) and waist–height ratio (WHTR) were then calculated and analyzed based on the results acquired. Metabolic syndrome was assessed in accordance to International Diabetes Federation [16]. Sociodemographic, medical anamnesis, anthropometric data and medical examination were performed in the Department of Geriatrics at the Medical University of Lodz, Poland. Secondly, subjects were referred to the Department of Hemostatic Disorders for further blood testing.

### 2.2. Basic Laboratory Measurements

The blood serum was assayed spectrophotometrically for fasting glucose concentration, total cholesterol (TC), low density lipoprotein cholesterol (LDL-C), high density lipoprotein cholesterol (HDL-C), triglycerides (TG), and uric acid (UA) (DIRUI CS 400, Changchun, China). Blood morphology was evaluated with 5-Diff Sysmex XS-1000i haematological analyser (Sysmex, Kobe, Japan). The concentration of homocysteine (Hcy) was measured using the analyzer Immulite 2000 XPi (Siemens, Erlargen, Germany). Plasma concentration of vWf, soluble VCAM-1 and soluble ICAM-1 were measured with commercially available ELISA kits provided by Abcam (Cambridge, UK). The determination of biochemical and morphological parameters has been previously described in the work of Karolczak et al. [17].

### 2.3. TAS, TOS, and Oxidation Markers Estimation

Total antioxidant status (TAS) and total oxidant status (TOS) in accumulated samples of blood plasma were evaluated with commercially available kits from LDN Labor Diagnostika Nord GmbH & Co. KG (Nordhorn, Germany). The parameters of antioxidant potential (TAS) were determined by the ABTS cation decay reaction [4,18]. The total content of lipid peroxides (the concentration of all lipid peroxides present in the tested samples, without distinguishing between individual molecules) was quantified in the sample, which was directly dependent on the concentration of free oxygen radicals. The determination of peroxides was carried out in the reaction of peroxidase-peroxides. The reaction converted tetra-methyl benzene (TMB) to a colored product. After the reaction was terminated, the reagent was photometrically measured (450 nm) [19,20].

Concentration of plasma protein and the protein content in platelet lysates were estimated with the Pierce™ BCA Protein Assay Kit (Thermo Fisher Scienctific, Waltham, MA, USA). Concentrations of free sulfhydryl groups in platelet and plasma proteins were assessed according to the methods reported earlier by Ando and Steiner [21]. The concentration of free amino groups in platelet and plasma proteins were evaluated according to Sashidhar et al. [22]. Appraisal of superoxide anion generation in blood platelets was performed by spectrophotometric method of cytochrome c reduction by superoxide anion, described by Gresele et al. [23]. The last reaction was conducted in two versions; with phosphate-buffered saline solution (PBS) and with Hcy. All of the abovementioned reactions have been described in detail in the Karolczak paper [24].

### 2.4. Statistical Analysis

The statistical analysis of the variables was performed on the basis of parametric and non-parametric tests. The analysis of empirical distributions of the variables was executed on the basis of the Shapiro–Wilk test. The employment of the parametric test type was also dependent on the homogeneity of variance and, therefore, the Levene test was used to verify this assumption. For non-parametric data, Mann–Whitney test was used. In order to describe the studied group, the structure indexes were calculated in the analysis of qualitative characteristics. To feature the mean value for the quantitative features, the arithmetic mean (m) and the median (Me) were calculated. The standard deviation (SD) was adopted as the measure of the dispersion, and the range of the variables was also specified, i.e., the values of the lower and upper quartiles.

When comparing two groups for discrete features (e.g., sex), the χ^2^ test of independence or the Fisher χ^2^ test are used. To check whether the observed associations may be different in men and women, a test of the homogeneity of slopes was performed for statistically significant relationships (after log-transformation of non-parametrically distributed data). Quantitative variables with a distribution deviating from the normal distribution were logarithmically transformed before being included in multivariate models. However, in the equations, they are presented in the non-transformed form. Generalized linear models were performed for the variables that were statistically significant in the univariate tests, as well as with all independent variables. Either Spearman’s rank correlation test (rho coefficient) or Pearson’s correlation test was used to assess the relationship between two quantitative variables. *p* < 0.05 was adopted as the lowest level of statistical significance. Statistical and graphical elaboration were implemented using the Statistica program 12.

## 3. Results

Table 1 presents the basic anthropometric data, cardiovascular risk variables, parameters of plasma antioxidant and oxidative capacity of 300 people. The table also includes the above-mentioned parameters for women and men, as well as a comparison of the respondents in the terms of sex.

Men were more likely to suffer from arterial hypertension and myocardial infarction and meet the criteria for metabolic syndrome (Table 2). In the case of hypercholesterolemia, type 2 diabetes, ischemic heart disease, stroke, obesity or smoking, there were no differences between men and women. There were also no differences in taking cardiovascular drugs.

Summary of correlations of basic cardiovascular risk factors with TAS and TOS markers and products of oxidative stress for the whole group are presented in Table 3 and separately for women and men in Table 4 and Table 5, respectively.

Metabolic syndrome, arterial hypertension, diabetes mellitus, myocardial infarction and chronic ischaemic heart disease were not related to TAS, TOS and oxidative markers. Subjects with diagnosed lipid disorders presented significantly lower concentration of platelet protein free thiols (Mann–Whitney z = −2.33 *p* = 0.01), platelet protein free amino groups (Mann–Whitney z = −2.01 *p* = 0.04), and superoxide anion generation under basal conditions or after in vitro homocysteinylation (Mann–Whitney z = −2.32 *p* = 0.01; z = −2.20 *p* = 0.02, respectively). Previous stroke was connected with significantly higher concentration of lipid peroxides in plasma (Mann–Whitney z = 1.91 *p* = 0.049). Smoking habit was associated with lower level of lipid peroxides in platelets (Mann–Whitney z = 2.20 *p* = 0.03). Obesity was related to higher TOS (Mann–Whitney z = 2.46 *p* = 0.01) (data not shown in the table).

Antiplatelet drugs usage was linked with lower level of free thiol and free amino groups of platelets proteins (Mann–Whitney z = −2.20 *p* = 0.02; z = −2.10 *p* = 0.03, respectively) and superoxide anion generation either by resting in or homocysteinylated platelets (Mann–Whitney z = −2.88 *p* = 0.003; z = −2.73 *p* = 0.006, respectively). Subjects taking angiotensin-converting enzyme inhibitors presented significantly lower concentration of free protein amino groups in plasma (Mann–Whitney z = −2.63 *p* = 0.008), and those who were treated with angiotensin II receptor blockers expressed lower TOS (Mann–Whitney z = −2.06 *p* = 0.03). Diuretics intake was related with significantly lower free protein thiol and amino groups concentration in plasma (Mann–Whitney z = −2.43 *p* = 0.01; z = −2.13 *p* = 0.03, respectively). Beta-adrenolytic drugs, Ca- blockers, hypolipidemic and antidiabetic drugs reveal no impact on TAS, TOS or oxidative markers. The variable “antihypertensive drugs” containing information about at least one antihypertensive drug used was related with significantly lower concentration of free thiol groups in plasma proteins (Mann–Whitney z = −2.60 *p* = 0.009) (data not shown in the table).

The results of tests of homogeneity of slopes revealed that there were no sex-related differences in the associations between cardiometabolic data and markers of redox balance. In the next step, the impact of age, sex, BMI, WHR, WHtR, cardiometabolic conditions, medications intake and other cardiovascular risk factors on TAS, TOS and other oxidative stress markers were consecutively analyzed. Generalized linear models were performed for variables statistically significant in univariate tests as well as with all independent variables gave identical results.

TAS in the univariate analysis was associated with TC and LDL-C (Table 3). In the generalized linear model only LDL-C affected the TAS.
TAS [mM] = 44.79 − 0.07 × LDL-C[mg/dL] (*p* = 0.002)(1)

In simple correlations, TOS reacted with BMI, WHtR, LDL-C and TG (Table 3). Moreover, women were characterized by higher TOS than men (Table 1). The usage of an angiotensin II receptor antagonist was associated with a lower TOS (Mann–Whitney z = −2.06; *p* = 0.03). Obesity was associated with a higher TOS (Mann–Whitney’s z = 2.46; *p* = 0.01). In the generalized liner model, TOS was influenced by BMI, sex and the usage of an angiotensin II receptor antagonist.
TOS [mM] = 0.01 × BMI [kg/m^2^] + 0.03 if woman + 0.07 if absence of an angiotensin II receptor antagonist (*p* < 0.001)(2)

Platelet lipid peroxides concentration in generalized linear model was associated only with vWF.
Plasma lipid peroxides [mmol/L] = −31.15 + 8.44 × vWF [µg/mL] (*p* < 0.001)(3)

Plasma lipid peroxides concentration expressed a positive correlation with ICAM-1 level. Furthermore, the concentration of plasma lipid peroxides was significantly higher among stroke survivors (Mann–Whitney z = 1.91; *p* = 0.049). Generalized linear model confirmed both ICAM-1 and previous stroke as independent statistical predictors.
Platelet lipid peroxides [nmol/μg of protein] = 0.008 × ICAM-1 [ng/mL] + 0.65 if past of stroke (*p* = 0.006)(4)

Concentration of free protein thiol groups in blood platelets was negatively related with uric acid and positively with vWF (Table 3). Subjects taking antiplatelet drugs presented a significantly lower level of free protein thiol groups in blood platelets (Mann–Whitney z = −2.20; *p* = 0.02), similarly to people diagnosed with lipid disorders (Mann–Whitney z = −2.33; *p* = 0.01). In the generalized linear model:Free thiol groups of platelet protein [μmol/μg of protein] = 12.18 × vWF [µg/mL] − 5.91 × uric acid [mg/dL] − 11.92 if present lipid disorders (*p* < 0.001)(5)

For plasma protein free thiol groups concentration in bivariable analysis, use of antihypertensive drugs was statistically significant, but only glucose was selected as an independent predictor in generalized linear model:Free thiol groups of plasma protein [μmol/μg of protein] = 0.0004 × glucose [mg/dL] (*p* = 0.001)(6)

For concentration of free amino groups of platelet proteins, only platelet count was selected:Free amino groups platelet proteins [nmol/μg of protein] = 8.46 − 0.02 × platelet count [10^3^/mm^3^] (*p* = 0.02)(7)

For plasma free protein amino groups level, only fasting glucose was selected:Free amino groups of plasma protein [mmol/mg of protein] = 0.25 × glucose[mg/dL] (*p* < 0.0001)(8)

Concentration of superoxide anion generated by resting blood platelets in bivariate analysis was positively associated with glucose. Furthermore, subjects taking antiplatelets drugs had significantly lower level of superoxide anion- (Mann–Whitney z = 2.88 *p* = 0.003). In the generalized linear model, superoxide anion was impacted only by glucose:Superoxide anion generated by resting platelets [1 × 10^8^ plt/mL dilutant] = −28.10 + 0.38 × glucose [mg/dL] (*p* < 0.001)(9)

Generation of superoxide by in vitro homocysteinylated blood platelets was positively linked with glucose level. Furthermore subjects taking antiplatelets drugs had a significantly decreased level of superoxide anion (Mann–Whitney z = 2.73 *p* = 0.006). In the generalized linear model, only glucose was selected:Superoxide anion generated by homocysteinylated platelets [1 × 10^8^ plt/mL dilutant] = −32.04 + 0.43 × glucose[mg/dL] (*p* < 0.001)(10)

## 4. Discussion

This is, to the best of our knowledge, the first study that analyses so profoundly the relationship between cardiometabolic diseases, risk factors and medications and the total antioxidant capacity of plasma, total oxidation potential and other non-routinely used markers of oxidative stress in the population of subjects aged 60–65 years.

### 4.1. TAS

In multivariate analysis, age, sex, anthropometric factors, cardiovascular disease, and medication use did not impact TAS. In terms of age, the presented result is consistent with some other studies on the antioxidant potential [25,26]. Some research has presented a decrease with age in total plasma antioxidant capacity [27,28]. It should be noted, however, that the study population was relatively homogeneous in terms of age, which could substantially affect the TAS value. The results did not reveal the relationship between TAS and obesity (estimated on the basis of BMI, WHR or WHtR). In the paper by Aslan et al., TAS was amplified with increasing BMI [29]. A similar outcome, indicating a positive correlation between BMI and TAS, was obtained in the study by Nälsén and co-authors, but only after taking into consideration the uric acid concentration [30]. Therefore, this association may be due to the influence of another factor combining TAS and obesity, such as uric acid or lipids. Previous research has presented numerous connections between markers of antioxidant potential and the development of ischaemic heart disease and the occurrence of myocardial infarction [31,32,33]. There are no such dependencies in our study, which may be associated with the observed low prevalence of cardiovascular diseases in the studied group.

The inclusion of classical cardiovascular risk factors in the multivariate analysis for TAS showed that TAS was dependent on LDL-C concentration. Higher LDL-C was associated with lower TAS. Antioxidants prevent the formation of ox-LDL-C during atherogenesis. LDL-C oxidation is inhibited by ascorbate or tocopherol, and lipid peroxidation is interrupted by vitamin A and β-carotene [34]. The consumption of antioxidants increases with elevated LDL-C levels. There are studies available in the literature indicating hypercholesterolemia as a parameter that may cause oxidative stress [35]. Research by Karabacak et al. proved that low concentration of HDL-C is associated with a significantly higher total oxidative potential, which, in turn, leads to an increased cardiovascular risk [36].

### 4.2. TOS

In the presented study, TOS was primarily associated with a higher BMI. It has been shown that obesity leads to increased parameters of oxidative stress, which in turn contributes to the development of obesity-related diseases [36]. Other studies also presented a positive association of BMI and oxidative stress [37,38,39]. In our study, women presented higher TOS. In Aslan and co-authors’ research, significantly higher TAS and TOS indicators were observed in the group of men [29]. Vassalle et al., on the other hand, showed significantly higher values of oxidative potential in the group of women [40]. Miller hypothesized that the lower TOS value in the group of women compared to men can be related to the protective effect of estrogens modulating the expression of NADPH oxidase and antioxidant enzymes. Most importantly, these studies were conducted in the group of women in the pre-menopausal period [36]. In another study, a higher TOS in the group of men with accompanying metabolic syndrome was shown by Kaya et al. [41]. Menopause and related hormonal disruptions may change the antioxidant–oxidative balance, which would explain the increased cardiovascular risk in the postmenopausal period. Interesting is the impact of angiotensin II receptor blocker on TOS in multivariable model. There is some research indicating that angiotensin II is a pro-inflammatory and pro-oxidative agent [42,43]. In the light of this assessment, our finding is consistent with previous conclusions. Consequently, the presented activity of the aforementioned group of antihypertensive medications may be considered an independent factor in lowering global cardiovascular risk.

### 4.3. Plasma and Platelet Lipid Peroxides

Plasma lipid peroxides concentration was assessed as a sum of all present lipid peroxides in the tested samples. In multivariable analysis, lipid peroxides are associated with ICAM-1 and a history of stroke. In terms of ICAM-1, the impact may be composite. Once we consider ICAM-1 as a potential cardiovascular risk factor, this is the reason to use ICAM-1 as coefficients in multivariable models. On the other hand, lipid peroxidation may induce endothelial activation, and increased levels of ICAM-1 circulating in blood plasma [44,45]. However, the basic role of ICAM-1 in the process of inflammation, such as activation and binding leukocytes to endothelium and further ROS production, cannot be forgotten [46]. At that point, ICAM-1 may be regarded as a factor for stimulation of lipid peroxidation. What is interesting is the presence of stroke in the patient’s history, which is consistent with a study on lipid peroxidation among subjects who had had strokes [47,48]. However, both studies referred to patients with a recently occurring ischemic incident. To conclude, the condition of elevated lipid peroxidation may be connected not only with acute phase of stroke, but regardless of treatment, may also be connected with a long-term condition. Higher platelet lipid peroxides were associated with higher vWF. The work of Koprivica et al. indicates the interrelation between oxidative stress markers and vWF, but only in the context of acute coronary disease [49]. Another study suggests that polymers of vWF under the process of peroxidation become hypofunctional, more pro-agglutinating, uncleavable to smaller forms, and, finally resistant to proteolysis by ADAMTS13 [50]. This enzyme also plays a role in minimalizing the damage caused by reactive oxygen species [51]. This may explain the question of higher vWF concentration caused by platelet lipid peroxides. Similarly, platelet lipid damage may lead to a greater release of platelets α-granules, which contain vWF [52].

For platelet lipid peroxidation, the glutathione peroxidase may be considered a one of the crucial enzymes [53]. This enzyme is supposed to have double function: via reduction of the lipid peroxides and thereby prevention of inactivation of nitric oxide, and via metabolism of nitric derivates of glutathione causing liberation of nitric oxide [54]. Although the activity of this enzyme has not been investigated in our study, it seems to be an important factor influencing redox balance and cardiovascular risk.

### 4.4. Plasma and Platelet Protein Free Thiol Groups

Protein oxidative changes disrupt the functioning of the cell. Oxidation of protein free thiol groups by superoxide radical, hydrogen peroxide, or hydroxide radical leads to biological inactivation of the protein, oxidation of membrane thiol groups, and disintegration of the cell barrier. In our study, the only parameter which independently influenced the plasma thiol groups is glucose. Van Dijk et al. indicated such a relation, but only in bivariate analysis. In a multivariable trial, the most prominent coefficient was the concentration of triglycerides [55]. Nevertheless, the cited work compared diabetic and non-diabetic participants and hyperglycemia seems to be most harmful factor for the plasma free SH group’s balance. Concentration of free protein SH groups of blood platelets in multivariable analysis is negatively impacted by concentration of UA and presence of lipid disorders and positively impacted by plasma concentration of vWF. The obtained results are consistent with the data presented by van Dijk et al. [55]. Still, the literature in this field is scarce, as is the information about the relationship between concentration of platelet protein free thiol groups and plasma concentration of vWF. Chen et al. conducted a study in which they proved that under oxidation, vWF was cleaved much slower by ADAMTS13 [56], and its accumulation provokes coagulation. The association between vWF and free -SH groups present in platelets may be explained by the fact that oxidation of vWF itself refers to thiol groups, so increased aggregation of this glycoprotein may be expected. Secondly, platelets in their granular bodies contain vWF in high concentration, which may additionally explain that relation. Still, this explanation is theoretical and needs further research. Finally, uric acid was found as a protective agent for thiol groups concentration. Uric acid is additionally treated as an antioxidant agent. In our previous study, we stated that uric acid is one of many factors decreasing platelet reactivity [24]. However, xanthine oxidase can play the role of oxidant enzyme that oxidizes thiol groups under certain stress conditions in the tissue [57]. The role of uric acid appears to be complex and multidirectional.

### 4.5. Plasma and Platelet Protein Free Amino Groups

Plasma proteins free amino groups in multivariable analysis were influenced only by accumulation of glucose. As is the case with the free thiol groups, hyperglycemia may play a detrimental role in the structure of plasma proteins. This may be proceeded by the glycation process of proteins with its outcome as glycation end products, which present oxidative and proinflammatory effect [58]. Advanced glycation end products formed after a non-enzymatic reaction between monosaccharides and free amino groups of proteins, lipids, and nucleic acids are responsible for a cascade of inflammation, oxidative stress, and disruption of calcium homeostasis [59]. The second factor affecting free amino groups is platelet count. Oxidative stress regulates the thrombotic process, including platelet activation and turnover [60].

### 4.6. Superoxide Anion Generated by Platelets

In the case of superoxide generation by blood platelets, either under basal conditions (platelets suspended in PBS) or after in vitro homocysteinylation, the glucose concentration was selected as an independent predictor. Prolonged hyperglycemia increases NOS gene expression, and consequently NO release. However, this up-regulation is associated with an increase in O_2_ production [61]. Other research has highlighted the effect of an increased level of glucose on mitochondrial dysfunction, and superoxide anion generation [62]. Antiplatelet agents (aspirin in vast majority), through their anti-inflammatory effects, attenuate the oxidative stress by nitric oxide, inhibit leukocyte and endothelium interaction [63], and enhance antioxidative pathways [64]. Therefore, both present and previous results unanimously indicate the fact that superoxide generation is closely related to modifiable factors such as glucose level.

### 4.7. Limitations of the study

The results of the study should be limited to relatively healthy Central-European adults aged 60–65 years. The observed associations may be different in diseased subjects, different age groups and different cultures. Another limiting aspect is related to the cross-sectional character of the presented study. Although they are relatively extensive, the obtained results give only some insight into complex redox-cardiovascular risks interactions. There are many other important factors influencing oxidative stress and redox balance, e.g., glutathione peroxidase.

## 5. Conclusions

To sum up, in this study, which is the first to make such a profoundly analysis of non-routinely used markers of oxidative stress, several interesting associations with cardiometabolic profiles have been observed. TAS, TOS and oxidative markers are largely related to modifiable cardiovascular risk factors such as body mass and intake of drugs such as angiotensin II receptor blockers. Plasma and platelet oxidation markers appear to be especially associated with glucose concentration. The presented analyses unanimously indicate that there are strong connections between cardiovascular risk factors and redox potential and specify how cardiometabolic interventions may counter-balance oxidative stress. Future prospective studies should corroborate those potential links between cardiometabolic profile and redox balance.

## Figures and Tables

**Table 1 antioxidants-11-01065-t001:** Characteristics in terms of cardiovascular risk factors and oxidative and antioxidant status by sex.

	*n* = 300	Females *n* = 150	Males *n* = 150
Age [years]	63 (61–64)	63 (61–64)	63 (62–64) * ^U^
Education [years]	13.0 (12.0–16.0)	13 (12–16)	13 (11–16)
BMI [kg/m²]	27.74 (24.98–30.92)	27.82 (24.51–31.22)	27.71 (25.30–30.55)
WHR	0.93 (0.84–1.0)	0.85 (0.80–0.91)	0.99 (0.95–1.03) *** ^U^
WHtR	0.57 (0.52–0.61)	0.56 (0.51–0.61)	0.58 (0.54–0.62) * ^U^
SBP [mmHg]	135 (125–149)	135 (123–145)	139.5 (125–151) * ^U^
DBP [mmHg]	82 (75–93)	80.5 (74–95))	85 (75–95) * ^U^
Pulse [/min]	67 (62–72)	67 (62–72)	67.0 (72.0)
Blood platelets [10^3^/mm^3^]	212 (181–244)	226.0 (200.0–266.0)	195.0 (165.0–226.0) *** ^U^
Total cholesterol [mg/dL]	203.5 (171.2	217.7 (180.8–249.6)	187.5 (165.8–218.6) *** ^U^
LDL cholesterol [mg/dL]	126.55 (100.7–154.8)	139.6 (103.4–168.0)	116.3 (97.4–141.5) *** ^U^
HDL cholesterol [mg/dL]	48.25 (40.9–57.6)	53.1 (45.7–64.2)	44.35 (38.7–51.6) *** ^U^
Triglicerides [mg/dL]	110.85 (77.8–159.8)	110.55 (76.6–159.4)	110.85 (78.4–162.15)
Glucose [mg/dL]	99.4 (91.4–110.8)	96.5 (89.9–107.6)	101.6 (93.7–114.4) ** ^U^
Uric acid [mg/dL]	4.8 (4.0–5.6)	4.2 (3.7–5.0)	5.4 (4.8–6.1) *** ^U^
Homocysteine [µmol/L]	14.5 (12.5–17.0)	14.15 (12.2–16.0)	15.3 (13.1–18.0) *** ^U^
vWF [µg/mL]	5.51(4.88–6.16)	5.42 (4.78–6.05)	5.56 (4.95–6.21)
VCAM-1 [ng/mL]	272.59 (249.18–298.13)	270.46 (247.04–292.81)	273.65 (249.17–304.51)
ICAM-1 [ng/mL]	210.05 (202.91–218.14)	210.04 (201.48–215.75)	210.05 (203.86–218.61)
TAS [mM]	41.6 (30.9–46.8)	40.99 (30.00–46.61)	42.07 (32.15–47.09)
TOS [mM]	0.54 (0.08–0.06)	0.55 (0.09–0.62)	0.52 (0.08–0.58) * ^U^
Plasma lipid peroxides [mmol/L]	0.27 (0.03–1.23)	0.30 (0.01–1.15)	0.25 (0.03–1.25)
Platelet lipid peroxides [nmol/μg of protein]	1.14 (0.47–24.35)	1.15 (0.41–25.58)	1.09 (0.05–23.76)
Free thiol groups of platelet protein [μmol/μg of protein]	2.84 (1.88–39.37)	2.83 (1.90–66.03)	2.83 (1.88–17.14)
Free thiol groups of plasma protein [μmol/μg of protein]	0.03 (0.02–0.05)	0.02 (0.02–0.04)	0.03 (0.02–0.04) * ^U^
Free amino groups of platelet protein [nmol/μg of protein]	0.15 (0.05–1.53)	0.14 (0.05–1.81)	0.16 (0.05–1.05)
Free amino groups of plasma protein [mmol/mg of protein]	16.79 (10.97–25.72)	16.51 (10.74–25.58)	16.95 (11.71–26.33)
Superoxide anion generated by resting platelets [1 × 10^8^ plt/mL dilutant]	0.37(0.13–3.50)	0.37 (0.11–3.51)	0.33 (0.13–3.28)
Superoxide anion generated by homocysteinylated platelets [1 × 10^8^ plt/mL dilutant] Hcy 1 × 10^8^ plt/mL dilutant	0.46 (0.16–4.55)	0.43 (0.14–4.02)	0.54 (0.17–4.54)

* *p* < 0.05; ** *p* < 0.01; *** *p* < 0.001; ***; * significantly different in comparison with the group of women. Data presented as mean ± SD or median (lower–upper quartile). Comparisons between men and women were performed with the use of the Mann–Whitney U test (^U^). Abbreviations: BMI: body mass index; WHR: waist–hip ratio; WHtR: waist–height ratio; SBP: systolic blood pressure; DBP: diastolic blood pressure; LDL: cholesterol—low density lipoprotein cholesterol; HDL: cholesterol—high density lipoprotein cholesterol; vWF: von Willebrand factor; VCAM-1: Vascular Cell Adhesion Protein 1; ICAM-1: Intracellular Adhesion Molecule 1; TAS: total antioxidant; TOS: total oxidative status.

**Table 2 antioxidants-11-01065-t002:** Cardiometabolic medical history and medication intake.

	*n* = 300	Females *n* = 150	Males *n* = 150
Metabolic syndrome	195	79	116 ***
Arterial hypertension	157	68	89 *
Hypercholesterolemia	198	105	93
Diabetes mellitus type 2	35	16	19
Myocardial infarction	15	2	13 **
Chronic ischaemic heart disease	43	17	26
Previous stroke	12	6	6
Smoking	69	31	38
Obesity (BMI ≥ 30 kg/m^2^)	95	47	48
Antiplatelets drugs	54	28	26
β-adrenolytic drugs	85	44	41
Ca-blockers	33	12	21
Angiotensin converting enzyme Inhibitors	70	33	37
Angiotensin II receptor blockers	27	17	10
Diuretics	57	29	28
Anihypertensive drugs (at least one)	137	70	67
Hipolipidemic drugs	69	28	41
Antidiabetic drugs	34	15	19

* *p* < 0.05; ** *p* < 0.01; *** *p* < 0.001; * significantly different in comparison with the group of women. Comparisons between men and women were performed with the use of the χ^2^ test or the Fisher χ^2^ test.

**Table 3 antioxidants-11-01065-t003:** Correlations for the whole testing group.

	TAS	TOS	Platelet Lipid Peroxides	Plasma Lipid Peroxides	Free Thiol Groups of Platelet Protein	Free Thiol Groups of Plasma Protein	Free Amino Groups of Platelet Protein	Free Amino Groups of Plasma Protein	Superoxide Anion Generated by Resting Platelets	Superoxide Anion Generated by Homocysteinylated Platelets
**Age**	−0.02	−0.07	−0.04	−0.05	0.00	−0.04	−0.01	−0.03	0.09	0.10
**Education status**	0.01	0.04	0.09	−0.01	0.12	−0.03	0.14 *	0.02	0.09	0.09
**BMI**	0.04	0.19 ***	0.04	0.05	−0.02	−0.12	0.03	−0.12	−0.03	−0.04
**WHR**	0.00	0.03	−0.12	0.07	−0.05	0.02	−0.02	0.03	−0.04	−0.05
**WHtR**	0.01	0.16 **	−0.06	0.06	−0.05	−0.08	0.03	−0.07	−0.04	−0.04
**SBP**	0.04	0.05	0.02	0.03	0.01	0.04	0.06	0.12	0.00	0.00
**DBP**	−0.02	0.01	−0.05	−0.01	−0.01	0.05	−0.01	0.06	0.02	0.01
**Number of blood platelets**	0.01	0.09	0.01	0.06	0.00	−0.07	−0.16 *	−0.10	−0.11	−0.11
**Total cholesterol (TC)**	−0.17 **	0.09	−0.01	0.03	−0.08	−0.07	−0.08	−0.02	0.03	0.03
**HDL cholesterol**	−0.01	−0.07	0.04	−0.08	−0.06	0.01	−0.11 *	0.02	0.07	0.07
**LDL cholesterol**	−0.16 *	0.12 *	−0.03	0.04	−0.08	−0.05	−0.10	−0.02	0.00	0.00
**Triglicerides**	−0.07	0.14 *	−0.04	0.07	−0.01	−0.05	0.10	−0.02	0.03	0.03
**Glucose**	−0.06	0.02	−0.10	−0.05	−0.01	0.13 *	0.12	0.13 *	0.19 **	0.21 *
**Uric acid**	0.11	0.08	−0.08	0.04	−0.13 *	0.03	−0.11	0.05	−0.07	−−0.08
**Homocysteine**	0.05	−0.06	0.02	−0.09	0.02	0.01	0.01	0.04	−0.05	−0.05
**vWF**	0.02	0.08	0.31 ***	−0.12	0.14 *	−0.10	0.08	0.10	0.04	0.05
**VCAM-1**	−0.01	−0.01	−0.06	−0.06	0.12	0.09	−0.10	0.03	−0.02	−0.02
**ICAM-1**	0.02	0.03	−0.11	0.19 *	−0.10	−0.05	−0.07	−0.08	−0.10	−0.10

* *p* < 0.05; ** *p* < 0.01; *** *p* < 0.001; The relationship between two quantitative variables was assessed by Spearman’s rank correlation test (rho coefficient) Abbreviations: BMI: body mass index; WHR: waist–hip ratio; WHtR: waist–height ratio; SBP: systolic blood pressure; DBP: diastolic blood pressure; LDL: cholesterol—low density lipoprotein cholesterol; HDL: cholesterol—high density lipoprotein cholesterol; vWF: von Willebrand factor; VCAM-1: Vascular Cell Adhesion Protein 1; ICAM-1: Intracellular Adhesion Molecule 1; TAS: total antioxidant status; TOS: total oxidative status.

**Table 4 antioxidants-11-01065-t004:** Correlations for the group of women.

	TAS	TOS	Platelet Lipid Peroxides	Plasma Lipid Peroxides	Free Thiol Groups of Platelet Protein	Free Thiol Groups of Plasma Protein	Free Amino Groups of Platelet Protein	Free Amino Groups of Plasma Protein	Superoxide Anion Generated by Resting Platelets	Superoxide Anion Generated by Homocysteinylated Platelets
**Age**	−0.03	−0.05	−0.01	−0.11	0.20 *	−0.08	0.14	−0.05	0.19 *	0.20 *
**Education status**	−0.02	0.06	0.00	−0.09	0.09	−0.09	0.04	0.04	−0.04	−0.04
**BMI**	0.14	0.14	0.06	0.17	−0.02	−0.17	0.08	−0.14	−0.02	−0.02
**WHR**	−0.05	0.17	−0.08	0.10	−0.11	−0.09	0.10	0.08	0.03	0.02
**WHtR**	0.07	0.15	−0.03	0.15	−0.05	−0.13	0.15	−0.06	0.03	0.03
**SBP**	0.04	0.23 **	0.02	0.26 **	−0.06	−0.09	−0.11	0.08	−0.09	−0.09
**DBP**	−0.08	0.18 *	−0.05	0.17 *	−0.11	−0.04	−0.17	−0.04	−0.10	−0.09
**Number of blood platelets**	0.03	0.18 *	−0.03	0.04	−0.17	0.00	−0.15	−0.12	−0.10	−0.10
**Total cholesterol (TC)**	−0.22 *	0.12	0.00	0.08	−0.10	0.01	−0.12	−0.01	0.02	0.03
**HDL cholesterol**	−0.01	−0.06	0.07	−0.04	−0.02	0.08	−0.15	0.03	−0.02	−0.02
**LDL cholesterol**	−0.26 **	0.12	−0.02	0.09	−0.13	0.02	−0.16	0.01	−0.02	−0.01
**Triglicerides**	0.00	0.12	−0.05	0.08	0.01	−0.09	0.21 *	−0.08	0.22 *	0.22 *
**Glucose**	−0.01	0.09	−0.06	−0.04	0.18	0.15	0.42 ***	0.10	0.47 ***	0.46 ***
**Uric acid**	0.17	0.17	−0.01	0.09	−0.21 *	−0.11	−0.14	−0.02	−0.14	−0.13
**Homocysteine**	−0.10	0.01	−0.04	−0.05	−0.07	−0.04	0.03	0.29 **	−0.05	−0.05
**vWF**	0.07	0.19 *	0.38 ***	−0.07	0.16	−0.09	0.07	0.21 *	0.07	0.07
**VCAM-1**	0.07	0.05	−0.10	−0.17	−0.12	0.08	−0.08	0.08	0.01	0.01
**ICAM-1**	0.11	0.15	−0.14	0.22*	−0.19 *	0.09	−0.12	0.06	−0.10	−0.10

* *p* < 0.05; ** *p* < 0.01; *** *p* < 0.001; The relationship between two quantitative variables was assessed by Spearman’s rank correlation test (rho coefficient)**.** Abbreviations: BMI: body mass index; WHR: waist–hip ratio; WHtR: waist–height ratio; SBP: systolic blood pressure; DBP: diastolic blood pressure; LDL: cholesterol—low density lipoprotein cholesterol; HDL: cholesterol—high density lipoprotein cholesterol; vWF: von Willebrand factor; VCAM-1: Vascular Cell Adhesion Protein 1; ICAM-1: Intracellular Adhesion Molecule 1; TAS: total antioxidant status; TOS: total oxidative status.

**Table 5 antioxidants-11-01065-t005:** Correlations for the group of men.

	TAS	TOS	Platelet Lipid Peroxides	Plasma Lipid Peroxides	Free Thiol Groups of Platelet Protein	Free Thiol Groups of Plasma Protein	Free Amino Groups of Platelet Protein	Free Amino Groups of Plasma Protein	Superoxide Anion Generated by Resting Platelets	Superoxide Anion Generated by Homocysteinylated Platelets
**Age**	−0.01	−0.08	−0.01	−0.03	−0.10	−.02	−0.10	0.01	0.00	0.01
**Education status**	0.01	0.03	0.16	0.02	0.12	0.04	0.19 *	0.01	0.17	0.17
**BMI**	−0.13	0.19 *	−0.04	−0.01	−0.04	−0.08	−0.03	−0.10	−0.04	−0.05
**WHR**	−0.09	0.10	−0.14	0.05	−0.08	−0.10	−0.20 *	−0.12	−0.18 *	−0.19 *
**WHtR**	−0.08	0.16	−0.11	0.01	−0.09	−0.08	−0.10	−0.11	−0.13	−0.14
**SBP**	0.00	−0.07	0.07	−0.07	0.05	0.09	0.17	0.16	0.06	0.06
**DBP**	−0.01	−0.13	0.02	−0.08	0.06	0.09	0.11	0.16	0.10	0.10
**Number of blood platelets**	−0.02	−0.03	0.05	0.15	0.14	−0.03	−0.14	−0.03	−0.10	−0.10
**Total cholesterol (TC)**	−0.05	0.07	−0.10	0.02	−0.06	−0.08	−0.04	−0.03	0.05	0.05
**HDL cholesterol**	−0.02	−0.17	−0.13	−0.08	−0.12	0.00	−0.12	0.01	0.16	0.17
**LDL cholesterol**	0.00	0.07	−0.08	0.03	−0.03	−0.07	−0.02	−0.04	0.04	0.04
**Triglicerides**	−0.12	0.17	−0.04	0.04	−0.03	−0.04	0.03	0.04	−0.09	−0.09
**Glucose**	−0.11	−0.02	−0.12	−0.06	−0.12	−0.10	−0.12	−0.07	−0.01	−0.02
**Uric acid**	0.01	0.15	−0.16	−0.02	−0.11	0.21 *	−0.14	0.10	−0.05	−0.06
**Homocysteine**	0.09	0.01	0.12	−0.16	0.06	0.00	−0.03	0.14	−0.06	−0.06
**vWF**	−0.05	−0.11	0.14	−0.10	0.11	−0.14	0.07	−0.12	0.00	0.01
**VCAM-1**	−0.07	−0.11	0.02	0.02	0.27 **	0.10	−0.10	−0.03	−0.04	−0.04
**ICAM-1**	−0.11	0.01	−0.02	0.14	−0.02	−0.24 **	−0.01	−0.37 ***	−0.11	−0.12

* *p* < 0.05; ** *p* < 0.01; *** *p* < 0.001; The relationship between two quantitative variables was assessed by Spearman’s rank correlation test (rho coefficient). Abbreviations: BMI: body mass index; WHR: waist–hip ratio; WHtR: waist–height ratio; SBP: systolic blood pressure; DBP: diastolic blood pressure; LDL: cholesterol—low density lipoprotein cholesterol, HDL: cholesterol—high density lipoprotein cholesterol; vWF: von Willebrand factor; VCAM-1: Vascular Cell Adhesion. Protein 1; ICAM-1: Intracellular Adhesion Molecule 1; TAS: total antioxidant status; TOS: total oxidative status.

## Data Availability

The statistical data used to support presented findings may be acquired by sending request to the corresponding author.

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
