# Peer review of "The Association of Oxidative and Antioxidant Potential with Cardiometabolic Risk Profile in the Group of 60- to 65-Year-Old Seniors from Central Poland"

_antioxidants, 2022, doi:10.3390/antiox11061065_

Round 1
Reviewer 1 Report
The manuscript by Soltyzic et al. reports on oxidative and antioxidant potentials in aging, in relation with cardiovascular risk, especially considering blood platelets in this risk. This is an interesting study performed on 300 people. Several comments can be made on some approximative measurements, especially relating with oxidative stress.
- In paragraph entitled Plasma and platelet lipid peroxides, on pp 13-14 (lines 372-394), the reader would like to know what peroxides have been measured. The Material & methods paragraph (pp 3-4 / lines 132-141) does not allow understanding the real markers which have been measured.
- One important enzyme controlling the peroxide tone in platelets is glutathione peroxidase (GPx). Unfortunately, GPx activity has not been investigated, and even not discussed.
- As a reference for lipid peroxidation studies in elderly people, the work by Vericel et al. (Thromb. Res. 1988) & Rey et al. (Biochim. Biophys. Acta 1994) should have been cited in both the Introduction and Discussion.
- As a minor comment, it could be underlined that the english form needs to be improved.
Author Response
The manuscript by Soltyzic et al. reports on oxidative and antioxidant potentials in aging, in relation with cardiovascular risk, especially considering blood platelets in this risk. This is an interesting study performed on 300 people. Several comments can be made on some approximative measurements, especially relating with oxidative stress.
- In paragraph entitled Plasma and platelet lipid peroxides, on pp 13-14 (lines 372-394), the reader would like to know what peroxides have been measured. The Material & methods paragraph (pp 3-4 / lines 132-141) does not allow understanding the real markers which have been measured.
Thank you for this comment. Unfortunately, we cannot determine what specific lipid peroxides have been measured. We could do this if we knew the molar absorption rates for the complexes of individual lipid peroxides with our chromogen. In our case, we examined the concentration of all lipid peroxides present in the tested samples, without distinguishing between individual molecules. We measured the global concentration of peroxides, because fractionation and the use of high-performance chromatographic methods were not available to us at that time. Yet, further explanation in that field was added to the manuscript.
- One important enzyme controlling the peroxide tone in platelets is glutathione peroxidase (GPx). Unfortunately, GPx activity has not been investigated, and even not discussed.
Thank you for this remark. We used already a big amount of data, also with non-routinely used markers of oxidative balance. We do agree, that the role of glutathione peroxidase is essential in redox balance, especially with regards to the molecular basis of atherosclerosis and platelet activity. Therefore, an additional sub-paragraph was added to the discussion.
- As a reference for lipid peroxidation studies in elderly people, the work by Vericel et al. (Thromb. Res. 1988) & Rey et al. (Biochim. Biophys. Acta 1994) should have been cited in both the Introduction and Discussion.
Thank you for the indication of crucial reference and citation papers. Both suggested papers were added to the manuscript.
- As a minor comment, it could be underlined that the english form needs to be improved.
The manuscript has been amended by a native speaker from foreign language section of the Medical University of Łódź.
Reviewer 2 Report
The manuscript written by Sołtysik and colleagues addresses an interesting topic. The study aims to to verify whether any non-previously examined element of oxide-93 reducing system might be of special link to any of cardiometabolic risk factors. There are some comments and one methodological aspect that authors should address.
Abstract
The abstract is lacking a conclusion. Please, add it.
I believe that, to maintain the title as it is, it should add at the end “in Warsaw”.
Background
The introduction is well written, however, it seems a little long to me. I suggest authors to shorten it.
The aim of the study is double:
- to estimate which of the most recognizable cardiovascular risk factors may play crucial role in the induction of oxide-reducing imbalance
- how metabolic syndrome and other cardiovascular disorders together with cardiometabolic drugs may modify this potential
I believe that the firs objective was addressed with the methodology and results exposed in the article, however, this is not the case for the second objective.
The design is cross-sectional, but in the aim it is stated that one parametermodifies another. Such information cannot be provided with a cross sectional study (which can only asses relations at the time of the measurements), but with longitudinal studies. I believe this is not the case, so I suggest authors to reconsider or rephrase the second aim, so it matches what it is presented in results.
Also, the second objective aims to see the link between metabolic syndrome and other CV disorders, plus drug intake. I only see association between factors or drugs with oxidative parameters, but not the sum of factors and drugs. If this is a redaction mistake, please correct. If not please provide such results. I am confused about the drug intake and how it was considered in the analysis. I cannot see the drug intake. Could you please clarify
Methods
Methodology section is well written. It is lacking information on whether this is a cross-sectional or longitudinal analysis.
Please provide further information on drugs and CV risk factors (see coment above)
Results
Results are well written. A big amount of data are presented.
Discussion
Discussion section is well written.
Are there any strengths in the present study? There is one limitation that is not indicated. Cross-sectional data can only provide associations, but not causality. I believe this is a limitation of the presented study.
The text in the discussion section is lacking a conclusion.
Author Response
Abstract
The abstract is lacking a conclusion. Please, add it.
According to the Reviewer’s suggestion this part of the abstract was reformulated.
I believe that, to maintain the title as it is, it should add at the end “in Warsaw”.
In fact, the presented study was performed in the city of Łódź, Central Poland. We added this information to the title.
Background
The introduction is well written, however, it seems a little long to me. I suggest authors to shorten it.
One of the paragraphs was removed from the paper.
The aim of the study is double:
- to estimate which of the most recognizable cardiovascular risk factors may play crucial role in the induction of oxide-reducing imbalance
- how metabolic syndrome and other cardiovascular disorders together with cardiometabolic drugs may modify this potential
I believe that the firs objective was addressed with the methodology and results exposed in the article, however, this is not the case for the second objective.
Thank you for this comment. In fact, the second aim was formulated imprecisely. Metabolic syndrome, cardiovascular disorders and cardiometabolic drugs were used in multivariate analyses to assess their potential contribution to oxide-reducing imbalance. Therefore, the objective was changed to be in line with the analyzes performed.
The design is cross-sectional, but in the aim it is stated that one parametermodifies another. Such information cannot be provided with a cross sectional study (which can only asses relations at the time of the measurements), but with longitudinal studies. I believe this is not the case, so I suggest authors to reconsider or rephrase the second aim, so it matches what it is presented in results.
As mentioned above, the second goal has been changed. The structure of cardiovascular diseases and the taken cardiometabolic drugs are presented in the Table 2. The vast majority of these variables were not significant in relation to the parameters of oxidative stress. Methodologically independent quantitative and qualitative variables were first checked for two-variable analysis – searching for the association with the parameters of oxidative stress. Only those variables that showed p <0.05 were included in the multivariate analysis. However, after taking into account the comments of the Reviewer, we re-analyzed the data – we performed a multivariate analysis taking into account all independent variables. Yet, such a procedure did not affect the results.
Also, the second objective aims to see the link between metabolic syndrome and other CV disorders, plus drug intake. I only see association between factors or drugs with oxidative parameters, but not the sum of factors and drugs. If this is a redaction mistake, please correct. If not please provide such results. I am confused about the drug intake and how it was considered in the analysis. I cannot see the drug intake. Could you please clarify
One of the most difficult elements to standardize was pharmacotherapy. Nowadays, antihypertensive, antidiabetic and antiplatelet therapy are diverse in terms of the substances used and their doses. In our study, drugs were divided into groups in a synthetic way. For example, anti-diabetic drugs included both oral drugs and insulin, because the numbers in each group were small. Only in the case of antihypertensive drugs, a broader division could be made, but this was due to the more frequent use of these drugs. In this case, the effect of all antihypertensive drugs at the same time wasn’t previously analyzed.
After analyzing the comments of the Reviewer, we came to the conclusion that such an approach may not, however, take into account all the associations. We performed an additional analysis for the variable "antihypertensive drugs (any)" containing information about at least one antihypertensive drug used. First, we checked whether this variable influences TAS, TOS and the analyzed elements of oxidative stress. It turned out that this variable statistically significantly influences only the free thiol groups in the plasma. Secondly, this variable was included in the multivariate analysis. However, in the generalized linear model it was already insignificant with the estimated value of p = 0.09.
The drug intake have been presented in the Table 2. The presentation of the results has also been modified to show the data more clearly.
Methods
Methodology section is well written. It is lacking information on whether this is a cross-sectional or longitudinal analysis.
Information about cross-sectional type of the study was added to the Methods section.
Please provide further information on drugs and CV risk factors (see coment above)
As mentioned above, the structure of cardiovascular diseases and the taken cardiometabolic drugs have been presented in the Table 2. Results has also been modified to show the data more clearly.
Results
Results are well written. A big amount of data are presented.
Thank you for your appreciation of gathered material.
Discussion
Discussion section is well written.
Are there any strengths in the present study? There is one limitation that is not indicated. Cross-sectional data can only provide associations, but not causality. I believe this is a limitation of the presented study.
This limitation has been added to the “Limitations” section. Any terms suggesting causality have been removed from the manuscript.
The text in the discussion section is lacking a conclusion.
Summary part was added to the discussion.
Round 2
Reviewer 2 Report
It wuould have been nice if authors had marked the changes on the manuscript.
I believe the article is suitable for publication.